# Application of High-Throughput Sequencing Technology in the Pathogen Identification of Diverse Infectious Diseases in Nephrology Departments

**DOI:** 10.3390/diagnostics12092128

**Published:** 2022-09-01

**Authors:** Yujuan Wang, Xiaoyi Hu, Lianhua Yang, Cheng Chen, Hui Cheng, Haiyun Hu, Wei Liang, Yongqing Tong, Ming Wang, Huiming Wang

**Affiliations:** 1Department of Nephrology, Renmin Hospital of Wuhan University, Jiefang Road 238, Wuchang District, Wuhan 430060, China; 2Department of Clinical Laboratory, Renmin Hospital of Wuhan University, Jiefang Road 238, Wuchang District, Wuhan 430060, China

**Keywords:** high-throughput sequencing, pathogen detection, urinary tract infection, peritoneal dialysis-associated peritonitis, central venous catheter related blood infections, lung infection

## Abstract

**Objective:** The purpose of this study was to explore the clinical applications of high-throughput sequencing (HTS) in the identification of pathogens in patients with urinary tract infection (UTI), peritoneal dialysis-associated peritonitis (PDAP), central venous catheter related blood infections (CRBIs), and lung infections in the nephrology department. **Methods:** Midstream urine samples from 112 patients with UTI, peritoneal fluid samples from 67 patients with PDAP, blood samples from 15 patients with CRBI, and sputum specimens from 53 patients with lung infection were collected. The HTS and ordinary culture methods were carried out in parallel to identify the pathogens in each sample. Pathogen detection positive rate and efficacy were compared between the two methods. **Results:** The pathogen positive detection rates of HTS in UTI, PDAP, CRBI, and lung infection were strikingly higher than those of the culture method (84.8% vs. 35.7, 71.6% vs. 23.9%, 75% vs. 46.7%, 84.9% vs. 5.7%, *p* < 0.05, respectively). HTS was superior to the culture method in the sensitivity of detecting bacteria, fungi, atypical pathogens, and mixed microorganisms in those infections. In patients who had empirically used antibiotics before the test being conducted, HTS still exhibited a considerably higher positive rate than the culture method (81.6% vs. 39.0%, 68.1% vs. 14.9%, 72.7% vs. 36.4%, 83.3% vs. 4.2%, *p* < 0.05, respectively). **Conclusion****s:** HTS is remarkably more efficient than the culture method in detecting pathogens in diverse infectious diseases in nephrology, and is particularly potential in identifying the pathogens that are unable to be identified by the common culture method, such as in cases of complex infection with specific pathogens or subclinical infection due to preemptive use of antibiotics.

## 1. Introduction

Infectious diseases remain the major challenge in nephrology departments [1,2]. They can emerge as a primary infection such as urinary tract infection (UTI), or present as a severe complication amid a certain chronic kidney disease such as peritoneal dialysis-associated peritonitis (PDAP) in peritoneal patients, central venous catheter related blood infections (CRBI) in hemodialysis patients, or lung infection in patients receiving glucocorticoid or immunosuppressive agent therapy. In each case, pathogen identification is critical, because it provides the basis for diagnosis and guidance for precise treatment. At present, culture detection is the most used method in clinical practice and is recommended in guidelines for some infectious diseases [3]. However, this traditional method has the inherent shortcomings of having low sensitivity and being time consuming, which lead to the low reliability and applicability in clinical practice. In the past decades, great efforts have been made to identify more efficient pathogen detection methods in the diagnosis of infectious diseases [4,5].

HTS can determine the DNA sequences by capturing the synthesized tags and can determine millions or even hundreds of millions of DNA or RNA sequences at the same time. HTS is unbiased, fast, and can theoretically detect all known pathogens [6]. HTS has been used for the diagnosis of bloodstream infections, nervous system infections, and respiratory infections by detecting pathogens’ DNA in blood, cerebrospinal fluid, and alveolar lavage fluid samples [7,8,9]. However, application of HTS in detecting infectious pathogens in patients with kidney diseases is rarely reported. This study explored the efficacy of HTS in detecting pathogens in diverse infections in nephrology, namely, UTI, PDAP, CRBI, and lung infection.

## 2. Materials and Methods

### 2.1. Patients

We reviewed patients that had been admitted to the Department of Nephrology, Renmin Hospital of Wuhan University, from August 2019 to January 2021, because of respective kidney disease with the indicated complications of UTI (n = 112, male: 26, female: 86, age: 14–92 with an average of 61.9 ± 14.8); PDAP (n = 67, male: 36, female: 31, age: 27–81 with an average age of 50.8 ± 14.5 years); CRBI (n = 15, male: 3, female: 12, age: 34–82 with an average of 57.87 ± 13.757 years); and lung infections (n = 53, male: 28, female: 25, age: 19–84 with an average age of 59.6 ± 17.5 years). The inclusion and exclusion criteria for the participants are listed at Table 1. The criteria ensured the presence of the indicated infection, and both pathogen detection methods were performed in parallel on each patient. The medical records of all participants were analyzed by the research team. This study was approved by the Ethics Committee of Renmin Hospital of Wuhan University (Approval Code: WDRY2020-k064; Approval Date: 25 February 2020). 

### 2.2. Sample Collection and Processing

Midstream urine samples were collected from patients with UTI. Peritoneal fluid samples were collected from patients with PDAP following the protocol recommended by ISPD [10]. Blood samples were obtained from three sites (peripheral vein and both catheter hubs) when a CRBI was suspected. Sputum samples were collected from patients with lung infections. All the samples were subjected to HTS and culture at the same time. Specimen cultivation and pathogen identification were conducted using automatic cultivation identification instrument and identification cards in accordance with the operating instructions [11] and reported to the clinicians after eliminating the possibility of contamination.

Samples were collected and subjected to the HTS to detect pathogens in the Laboratory of Renmin Hospital of Wuhan University. The target pathogens for HTS identification included bacteria, fungi, mycoplasma, chlamydia, rickettsia, tuberculosis, and other atypical pathogens. Detection procedures have been described elsewhere [12], and briefly comprised the following steps: (1) sample processing and nucleic acid extraction; we used the Sansure DNA Extraction Kit (Changsha, China); (2) amplification and nanopore targeted sequencing; NTS was built by targeted amplification of the 16s rRNA gene (for bacteria), IST1/2 gene (for fungi), and rpoB (for Mycobacterium spp.) using universal and specific primers, and sequenced by a real-time nanopore sequencing platform; (3) data analysis: sequencing data were divided into samples according to the sequencing tags, adapters were removed, low-quality sequences were filtered, host sequences were removed based on the BLAST and pathogens’ databases, and sequences were annotated; (4) potential pathogen determination: closely related microorganisms were filtered out, contaminants from NTS laboratory sampling and from human normal flora were filtered out by negative controls [13], and a reportable list of clinical pathogens referenced in published case reports was used [14].

### 2.3. Statistical Analysis

SPSS 22.0 software was used for statistical analysis. Enumeration data are expressed as number of cases and rate. Paired samples were compared by the chi-square test or the Fisher exact probability method and consistency test (McNemar test and Kappa test). *p* < 0.05 was considered statistically significant. 

## 3. Results

### 3.1. Comparison of the Detection Positive Rate between the Two Methods

In 112 patients with UTI, potential pathogens were identified in 95 cases (84.8%) by HTS, compared with 40 cases (35.7%) by the culture method (*p* < 0.001). Among the positive detections, there were nine cases with complete consistency and 21 cases with partial consistent between two test methods. Partial consistency was defined as cases of shared pathogens detected by both methods on the same sample, in which the other pathogen was only detected by one approach. Eight cases showed totally distinct test results between HTS and the culture method. Of note, two cases with negative findings by methods of HTS detection displayed a positive test result by the culture method. In 67 patients with PDAP, potential pathogens were identified in 48 cases (71.6%) by HTS, compared with 16 cases (23.9%) by the culture method (*p* < 0.001). The positive test results in nine cases were completely consistent in two methods and in three cases were partially consistent. In 15 patients with CRBI, potential pathogens were identified in 12 cases (75%) by HTS and in seven cases (46.7%) by the culture method (*p* < 0.001). Among these, results in four cases were totally consistent in the two methods, and in one case were partially consistent. In 53 patients with lung infection, potential pathogens were identified in 45 cases (84.9%) by HTS, strikingly higher than the three cases (5.7%) by the culture method (*p* < 0.001). One case showed fully consistent results in the two methods and two cases were partially consistent. The comparison of the positive rate of detection between HTS and the culture method is shown in Table 2 and Figure 1A. Figure 1B shows the profile of detection results of the two methods in each infectious disease.

### 3.2. Profile of Identified Pathogens by HTS and Culture Methods on Indicated Diseases

The profiles of the identified pathogens in the two approaches for the infectious diseases were further analyzed in terms of the strain number and their species. As shown in Appendix A, HTS detected a total of 211 strains (consisting of 166 bacterial strains, 34 fungal strains, 7 *Ureaplasm* strains, and 4 *Mycoplasma* strains), compared with a total of 43 strains (consisting of 34 bacterial strains and 9 fungal strains) detected by the culture method in 112 UTI patients. In 67 patients with PDAP, HTS produced 68 strains of potential pathogens including 52 bacterial strains, 13 fungal strain, 2 *Mycobacterium tuberculosis* strains, and 1 rickettsia, and the culture method yielded only 16 strains including 13 strains of bacteria and 3 strains of fungi. In 15 patients with CRBI, a total of 22 strains of potential pathogens (52 bacterial strains, 13 fungal strain, 2 *Mycobacterium tuberculosis* strains, and 1 rickettsia) were detected by HTS, in sharp contrast to 7 strains being detected by the culture method. Similarly, HTS identified 111 potential pathogen strains with species of bacteria, fungi, *Mycoplasma orale*, *Cryptosporidium*, and *Pneumocystis jiroveci* in 53 patients with lung infection, overwhelmingly higher than that the 3 strains identified by the culture method. These results strongly suggest that HTS is superior to the culture method in terms of detection success regardless of the pathogen type. More detailed information about the identified potential pathogens linked to the indicated diseases or detection approaches are summarized in Appendix A.

### 3.3. Comparison of the Detection Effectiveness by Pathogen Type between HTS and Culture Methods

The detection effectiveness of the culture method is largely limited by the pathogen type and, in particular, its environmental dependent viability. On the contrary, due to the approach used in HTS, its high detection effectiveness is independent of pathogen type. We then compared the detection effectiveness by pathogen type between HTS and culture methods for four infectious diseases. In this part of the study, the microorganisms detected from a patient’s specimen, regardless of the detection method adopted, are recognized as the cause of the patient’s infection. Therefore, in these subjects, some patients are infected by single pathogenic microorganisms, and some patients may have mixed infection. The detected microorganisms were grouped into the four categories of bacteria, fungi, atypical pathogens, and parasites. Atypical pathogenic microorganisms refer to mycoplasma, chlamydia, or mycobacterium tuberculosis. We found that pathogens of bacteria and fungi were detected in CRBI patients; bacteria, fungi, and atypical pathogens were detected in UTI patients and PDAP patients; and bacteria, fungi, atypical pathogens, and parasites were detected in lung infection patients. HTS produced a considerably higher positive test rate for each pathogen type than the culture method in UTI patients (for bacteria 97.78% vs. 36.67%, *p* < 0.00; for fungi 97.14% vs. 25.71%, *p* < 0.001; for atypical pathogens 100% vs. 0, *p* < 0.001), in PDAP patients (for bacteria 93.02% vs. 32.23%, *p* < 0.001; for fungi 100% vs. 20.10%, *p* < 0.001; for atypical pathogens 1 of 1 patient vs. 0 of 1 patient), in CRBI patients (for bacteria 100% vs. 58.30%, *p* < 0.001; for fungi 3 of 3 patients vs. 0 of 3 patients), and in lung infection patients (for bacteria 100% vs. 7.32%, *p* < 0.001; for fungi 100% vs. 0, *p* < 0.001; for atypical pathogens 1 of 1 patient vs. 0 of 1 patient; for parasites 1 of 1 patient vs. 0 of 1 patient) (Table 3). These results demonstrated that the HTS method possesses a higher positive detection positive rate overall than the culture method by pathogen type. This advantage is especially obvious for fungi, atypical pathogenic microorganisms, and parasites. 

### 3.4. Different Impacts of Prior Use of Antibiotics on the Detection Efficiency of the Two Methods

A total of 76 of 112 patients with UTI, 47 of 67 patients with PDAP, 11 of 15 patients with CRBI, and 48 of 53 patients with lung infection received empirical antibiotic therapy before the pathogen tests. For the culture method, the positive rate in patients with prior use of antibiotics was lower than that in patients with no prior use of antibiotics, but the difference was not statistically significant (27.8% vs. 39.5% in UTI patients, *p* = 0.228; 14.9% vs. 45.0% in PDAP patients, *p* = 0.056; 36.4% vs. 50.0% in CRBI patients, *p* = 0.634; 4.2% vs. 20.0% in lung infection patients, *p* = 0.145). Although the HTS methods in patients with prior use of antibiotics also showed an inferior positive rate than that in patients with no prior use of antibiotics, the difference was still not statistically significant (81.6% vs. 91.7% in UTI patients, *p* = 0.165; 68.1% vs. 80% in PDAP patients, *p* = 0.322; 72.7% vs. 100% in CRBI patients, *p* = 0.243; 83.3% vs. 100% in lung infection patients, *p* = 0.321). However, the HTS method had a better positive detection rate than the culture method regardless of prior antibiotic use, especially in patients with prior use of antibiotics (81.6% vs. 27.8% in UTI patients, *p* < 0.001; 68.1% vs. 14.9% in PDAP patients, *p* < 0.001; 72.7% vs. 36.4% in CRBI patients, *p* < 0.001; 83.3% vs. 4.2% in lung infection patients, *p* < 0.001) (Table 4).

## 4. Discussion

Infectious diseases, either present as complications or comorbidities, are frequently seen in patients with kidney diseases due to their impaired immunity and invasive operations. UTI, PDAP, CRBI, and lung infection are the major infection diseases encountered by nephrologists in a nephrology department. The full diagnosis of infection, including pathogen identification, is essential in guiding treatment. Traditionally, the culture method has been widely implemented in clinics and is recognized as the gold standard for infection diagnosis. However, the culture method is far from promising because it is time consuming and has relatively low detection efficiency. These are particular challenges when the pathogen load in the specimen is scarce or the activity is inhibited; for instance, if the patient received empirical antibiotics therapy before taking the test. HTS has emerged as a powerful approach in molecular diagnosis with high effectiveness and accuracy [15,16,17,18]. Recently, HTS has been applied in in the diagnosis of bloodstream, nervous system, and respiratory tract infections by detecting the pathogens in various samples, such as blood, cerebrospinal fluid, and alveolar lavage fluid specimens [19,20,21]. It has been demonstrated that the HTS method can detect a large variety of pathogens, such as bacteria, fungi, viruses, and parasites, with the advantages of high sensitivity, high throughput, requiring less time, and being less dependent on patients’ conditions.

In this study, we retrospectively evaluated the effectiveness of HTS in the diagnosis of diverse infectious diseases in the Department of Nephrology. A total of 112 patients with UTI, 67 patients with PDAP, 15 patients with CRBI, and 53 patients with lung infections were included and the McNemar test was used to compare the pathogen detection rates by HTS and culture methods. The results showed that the HTS method has a significantly higher positive detection rate than the culture method. HTS can detect potential infectious pathogens in most of the culture-negative specimens. However, among patients with UTI, there were two cases in which HTS did not detect the pathogen, while the culture method detected *Escherichia coli* and *Citrobacter freundii*. In three cases of PDAP, pathogens were not detected by HTS, but *Escherichia coli*, *Enterococcus faecalis,* and *Staphylococcus aureus* were detected by the culture method. These results indicate the possibility of false and missed detection by the HTS method.

Our results showed that the potential pathogens of UTIs detected by HTS were mainly Gram-negative bacteria (48.4%), followed by Gram-positive bacteria (30.3%) and fungi (16.1%). The most common potential pathogens were *Escherichia coli*, *Enterococcus faecium*, *Gardnerella vaginalis,* and *Enterococcus faecalis*, which were consistent with those reported in the literature [22]. The potential pathogens of PDAP detected by HTS were mainly Gram-positive bacteria (42.6%), followed by Gram-negative bacteria (33.8%) and fungi (19.1%). Common potential pathogens include *Streptococcus*, *Escherichia coli*, *Aspergillus*, *Staphylococcus epidermidis*, and *Pseudomonas aeruginosa*, which were consistent with those reported in the literature [23,24]. In CRBI, the potential pathogens detected by HTS were mainly Gram-positive bacteria (63.6%), followed by Gram-negative bacteria (22.7%) and fungi (13.6%). The most common causative micro-organisms were *Staphylococcus aureus*, *Corynebacterium*, and *Anaerococcus*, which were consistent with those reported in the literature [25,26,27,28]. In lung infections, the potential pathogens detected by HTS were mainly Gram-positive bacteria (54.9%), followed by fungi (21.6%) and Gram-negative bacteria (20.7%). The most common pathogens were *Streptococcus*, *Candida albicans*, *Rosella glutinosa*, and *Neisseria*. These findings were not consistent with previous reports that common respiratory pathogens were predominantly Gram-negative bacteria, with *Pseudomonas aeruginosa*, *Streptococcus pneumonia*, and *Klebsiella pneumoniae* being the most common pathogens [29,30]. A possible reason for this is the small size of the patients with lung infection included in this study, and the fact that the patients had combined kidney disease and were immune-compromised.

In this study we evaluated the detection efficiency of HTS and culture methods by the identified pathogen type of bacterial, fungal, parasite and atypical pathogens. The overall detection rate of the HTS method was higher than that of the culture method, and HTS was superior to the culture method in the diagnosis of bacteria, fungi, atypical pathogens, and mixed infections. Furthermore, HTS was superior to the culture method in detecting fungi, atypical pathogens, anaerobic bacteria (*Prevotella*, *Fingoldi*, *Micromonas*, etc.), and aerobic bacteria *(Streptococcus*) that require strict culture conditions. These pathogens are difficult to detect under regular microbiological laboratory culture conditions, whereas HTS is an unbiased method that does not rely on culture conditions. In theory, HTS can detect all types of pathogens in clinical samples except viruses [31]. 

Etiology-based prescription of antibiotics is advocated in clinical practice to avoid drug resistance and unnecessary medical costs caused by antibiotic abuse. However, in some conditions, empirical use of antibiotic before a pathogen detection test is permitted. For these patients, the follow-up pathogen identification is still critical for late drug adjustment, so such identification cannot be abandoned. When the traditional culture method is used to detect the etiology of these patients, its positive rate is usually very low and its clinical value is limited. In these circumstances, HTS can be used as an effective alternative detection method. Our results showed that, in patients having used antibiotics, the pathogen positive detection rate of HTS is significantly higher than that of the ordinary culture method, indicating that HTS is less affected by prior use of antibiotics. The high detection rate of HTS may be ascribed to the fact that pathogens’ DNA exists in the plasm for a longer time and its detection is unlikely to be influenced by antibiotics [17]. 

In summary, HTS has pronounced advantages in the diagnosis of diverse infectious diseases (urinary tract infections, peritoneal dialysis-related peritonitis, and lung infections) in nephrology departments, especially for fungi, atypical pathogens, parasites, and mixed infections. Precise treatment based on detection of pathogens can be applied, thus reducing unnecessary and inappropriate antibiotic use and avoiding clinical abuses of antibiotics. In addition, HTS is less time-consuming and more efficient than the culture method. For critically ill patients with infections, the use of HTS to detect pathogens is of greater significance and can improve the early detection of pathogens. HTS can facilitate the accurate diagnosis and precise treatment of patients with infectious diseases, shorten the length of hospitalization, and reduce the mortality rate. HTS can be used as an effective supplement to the traditional culture method. The combination of both methods can improve the overall pathogen detection rate and benefit the patients.

## Figures and Tables

**Figure 1 diagnostics-12-02128-f001:**
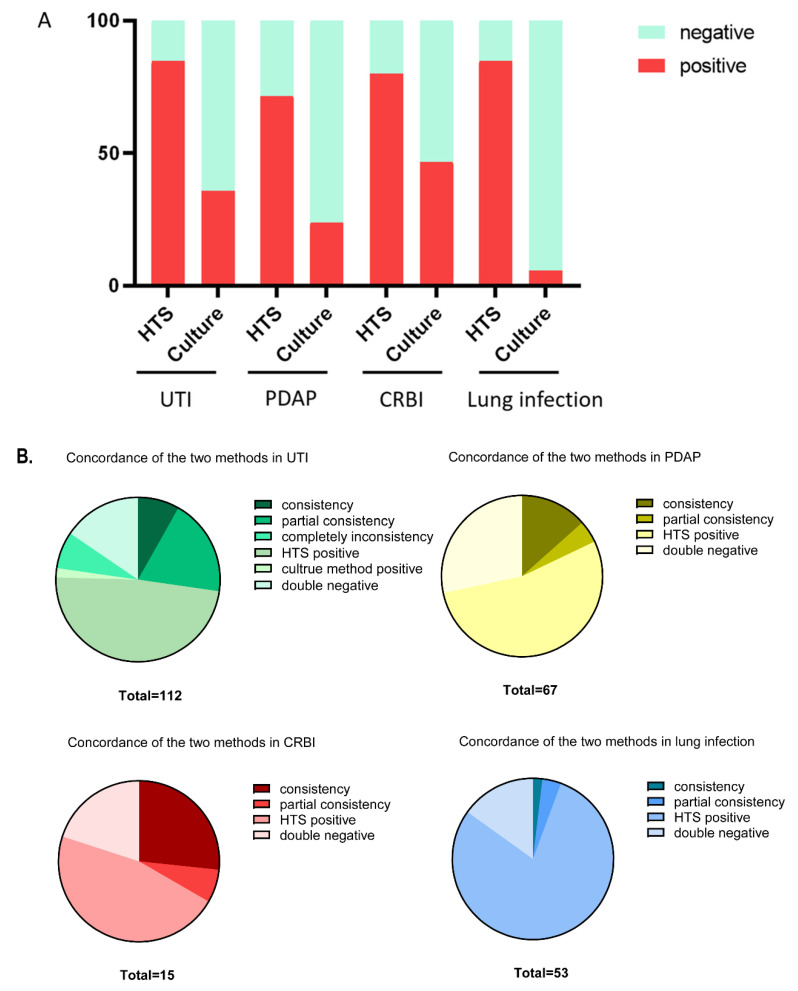
Profile of the detection results using HTS and culture methods on indicated infectious diseases. (**A**) Comparison of positive rate of detection between HTS and culture method. (**B**) Profile of detection results of the two methods in each infectious disease.

**Table 1 diagnostics-12-02128-t001:** Criteria for diagnosis and exclusion.

	Diagnostic Criteria	Exclusion Criteria
UTI	(1) Patients without urinary catheters: clinical manifestations, urinary leukocyte (WBC) count (≥10^4^ CFU/mL, leukocyte count with Sysmex UF-1000i urinary sediment analyzer), pathogenic bacteria ≤2 kinds, and ≥10^3^ CFU/m (2) Patients with catheterization: clinical manifestations, pathogens ≤2, and ≥10^5^ CFU/mL, regardless of leukocyte count. The clinical manifestations of upper urinary tract infection include renal pain and fever, while the clinical manifestations of lower urinary tract infection include frequent micturition, urgency, and pain	(1) Incomplete clinical data (2) Only one of urine culture or HTS was performed
PDAP	Peritonitis can be diagnosed in peritoneal dialysis patients with more than 2 of the following 3 items: (1) Abdominal pain, peritoneal exudate turbidity, with or without fever (2) Leukocyte count in penetrant >10 × 10^7^/L, neutrophil ratio >50% (3) The culture of pathogenic microorganisms in penetrant was positive	(1) Patients with other types of abdominal or pelvic infection (2) Patients with incomplete clinical data (3) The time of starting peritoneal dialysis was less than 1 month (4) Peritoneal dialysis combined with hemodialysis (5) Only one of the tests of peritoneal dialysis effluent culture or HTS was performed
CRBI	(1) The patients on in-center hemodialysis using catheters displayed fever, chills, rigors, hypotension before or during the hemodialysis session (2) New unexplained malaise, with concurrent exclusion of catheter–unrelated infectious foci	(1) Incomplete clinical data (2) Only one of culture or HTS was performed
Lung infection	(1) Recent cough, expectoration, or aggravation of original respiratory disease symptoms, with or without purulent sputum, chest pain, dyspnea, and hemoptysis (2) Fever (3) Signs of pulmonary consolidation and/or wet rales (4) WBC of peripheral blood leukocytes >10 × 10^9^/L or <4 × 10^9^/L, with or without nuclear left shift (5) Chest imaging examination showed new patchy infiltration, leaf or segment consolidation, ground glass shadow or interstitial changes, with or without pleural effusion. Meet one of item (5) and other items, except pulmonary tuberculosis, pulmonary tumor, non-infectious pulmonary interstitial disease, pulmonary edema, atelectasis, pulmonary embolism, pulmonary eosinophilic infiltration and pulmonary vasculitis	(1) Incomplete clinical data (2) Only one of culture or HTS was performed

**Table 2 diagnostics-12-02128-t002:** Comparison of positive rate of detection between HTS and culture methods.

Category	HTS			Culture Method			*p*
	Positive	Negative	Detection Rate	Positive	Negative	Detection Rate	
UTI	95	17	84.8%	40	72	35.7%	<0.001
PDAP	48	19	71.6%	16	51	23.9%	<0.001
CRBI	12	3	75.0%	7	8	46.7%	<0.001
Lung infection	45	8	84.9%	3	50	5.7%	<0.001

**Table 3 diagnostics-12-02128-t003:** Comparison of the effectiveness between high-throughput sequencing and the culture method in detecting pathogens.

	Diseases	Bacterial			Fungi			Atypical Pathogen			Parasite		
Pathogens		Positive Rate of HTS	Positive Rate of Culture Method	*p*	Positive Rate of HTS	Positive Rate of Culture Method	*p*	Positive Rate of HTS	Positive Rate of Culture Method	*p*	Positive Rate of HTS	Positive Rate of Culture Method	*p*
UTI		97.78%	36.67%	<0.001	97.14%	25.71%	<0.001	100%	0	-	-	-	-
PDAP		93.02%	30.23%	<0.001	100%	23.10%	<0.001	100%	0	-	-	-	-
CRBI		100%	58.30%	<0.001	100%	0	-	-	-	-	-	-	-
Lung infection		100%	7.32%	<0.001	100%	0	-	100%	0	-	100%	0	-

Note: The positive rate of the indicated method for the indicated species was calculated by dividing the number of positive samples detected by HTS or culture method by the number of total positive samples detected by the two methods as the denominator.

**Table 4 diagnostics-12-02128-t004:** Impacts of prior use of antibiotics on the detection efficiency of the two methods.

	Detection Rates	Prior Use of Antibiotics		No Prior Use of Antibiotics		*p*
Diseases		HTS	Culture-Based Method	HTS	Culture-Based Method	
UTI		81.6%	27.8%	91.7%	39.5%	0.228 *; 0.165 ^#^; 0.000 ^△^; 0.026 ^☆^
PDAP		68.1%	14.9%	80.0%	45.0%	0.056 *; 0.322 ^#^; 0.000 ^△^; 0.043 ^☆^
CRBI		72.7%	36.4%	100.0%	50.0%	0.634 *; 0.243 ^#^; 0.000 ^△^; 0.048 ^☆^
Lung infection		83.3%	4.2%	100.0%	20.0%	0.145 *; 0.321 ^#^; 0.000 ^△^; 0.048 ^☆^

*: *p*-value of the positive rate in patients with prior use of antibiotics vs. the positive rate in patients with no prior use of antibiotics in culture method; ^#^: *p*-value of the positive rate in patients with prior use of antibiotics vs. the positive rate in patients with no prior use of antibiotics in HTS method; ^△^: *p*-value of the positive rate of HTS method vs. culture method in patients with prior use of antibiotics; ^☆^: *p*-value of the positive rate of HTS method vs. culture method in patients with no prior use of antibiotics.

## Data Availability

Data is contained within the article or Appendix A.

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
