# Peer review of "Application of High-Throughput Sequencing Technology in the Pathogen Identification of Diverse Infectious Diseases in Nephrology Departments"

_diagnostics, 2022, doi:10.3390/diagnostics12092128_

Round 1

Reviewer 1 Report

Wang et al present very interesting data on alternative diagnostic approaches to potential infectious disease cases in a hospital-based nephrology unit. They compare the use of culture-based diagnosis with the evolving technology of targeted nanopore sequencing, which is much faster and more amenable to infectious disease clinical diagnosis than any DNA sequencing approach yet developed. I found the data presented to be fascinating - the identities and ranges of microbes detected in each context, and the impact of prior antibiotic use on detection frequency. 

The greatest concern I have about this work is that the authors appear to uncritically accept that every microbe detected in the sequencing data is an actual pathogen. (Only one time is the phrase “potential pathogen” used in the entire manuscript.) In fact, many of the microbes detected are known commensals of skin, gastrointestinal tract, or genitourinary tract, and thus could feasibly be either sample contaminants or sequelae of the actual infection, and not responsible for pathology. 

Nanopore targeted sequencing, particularly since it includes an amplification step, is potentially very sensitive, which is both a blessing and a curse. This is a new enough approach that in most situations - sample, source, and microbe involved - there are no standardized criteria for deciding what represents a true infection with potential pathological outcomes for the patient. 

Diagnosticians and clinicians must be very cautious about assigning pathogenicity to microbes showing up in the sequence data. For example, looking at UTI data in Supplementary Table 1, there are many species in the HTS data that are well established vaginal microbiota (e.g. Lactobacillus, Bifidobacterium, Peptoniphilis), not generally thought to be associated with UTI. Urine is certainly susceptible to contamination by these species, probably more so than the other specimens presented in this work. The bottom line is that the authors need to be very clear that the appearance of DNA from a particular microbe, whether bacterial or fungal, in a specimen does not automatically implicate that organism in the clinical condition being examined. I’m reluctant to recommend publication of this manuscript without qualifying statements regarding this issue - one simply cannot take at face value that detection of a sequence in a specimen identifies that microbe as being A or THE pathogen. 

Presumably the clinicians involved in these cases did not base their treatment decisions solely on sequencing results. It would be useful to readers for the authors to include in the discussion section something about how such data were actually used as a component of the clinical assessment of patients, as this is an evolving field. It would also be useful for the authors to discuss more the situations in which the sequence data did NOT agree with the culture data - how was that resolved? That dilemma illustrates a crucial point in the application of this new approach to infectious disease diagnosis - what approaches ARE possible to assess the validity and relevance of this type of sequence data?  

There are several other specific problems in the manuscript that can easily be addressed by the authors. 

  1. The authors frequently use the term “strain” when they mean “species”. In microbiology (or biology in general), “strain” is applied to genetic variants WITHIN a species. The authors here are intending to refer to different species. And in Supplemental figures 1-4, the X axis label should replace “Number of strains” with “Number of isolates”.)

  2. How the authors define the “positive rate of HTS” in Table 3 is unclear. For example, on line 110-12 (first paragraph of results), it says that among 67 PDAP samples, “pathogens” were identified in 48 (71.6%) by HTS, and 16 by culture. However, Table 3 indicates a “positive rate of HTS” of 93% for bacterial pathogens, 100% for fungi, and 100% for atypical pathogens. What is the denominator for those calculations? What independent measure of the presence of those pathogens are the authors comparing HTS and culture to? As far as I can tell, there are only two measures of the presence of pathogens being used here - HTS and culture, so they can only be compared to each other, or to the total number of samples.

  3. Line 246 - The authors note that the commonly detected lung “pathogens” are “not consistent with previous reports” - what isn’t consistent, and why? 

  4. Line 257 - The sequencing approach used in this work (nanopore targeted sequencing, which is based on amplification and sequencing of bacterial 16S rRNA sequences, along with fungal internal spacer sequences) cannot identify “all types of pathogens” - it will explicitly NOT be able to detect viral pathogens. Only a metagenomic approach will detect viruses. 

  5. Supplemental Table 1 - the right side is completely cut off from the PDF so that the lung data isn’t shown. Also in this table, there are numerous typographical errors and misspellings. For example, “Fingoldi magna” should actually be Finegoldia magna. Staphylococcus epidermidis shows up twice, once as “Staphylococcusepidermidis”. There are numerous other errors, and I don’t know what to make of “dystrophic bacteria” and “hemolytic twins” as “identified pathogens”. Please go through this table carefully.

Author Response

We appreciate the insightful comments and valuable advice offered by the reviewer!

The reviewer #1 raised the major concern regarding the validation of the detected microbes in the sequencing data. It seemed to be less critical to accept that every microbe detected in the sequencing data was an actual pathogen without taking into account contamination or false positive results. This is a key issue always arose concerns and debates and needed to be evaluated carefully. To avoid false positive detection of pathogen by NGS method, the Department of Clinical Laboratory of our hospital made the following efforts before issuing the HTS report:

  • To monitor the contamination introduced in the laboratory of the NTS test, we used two extraction controls (Tris-EDTA buffer with DNA extraction processing) and two no-template controls (Tris-EDTA buffer) were batched in one sequencing library. To filter out contaminants from human normal flora, we used samples from healthy individuals. The detailed logic of filtering potential laboratory contaminants were performed based on previously reported method (Jing C, Chen H, Liang Y, Zhong Y, Wang Q, Li L, Sun S, Guo Y, Wang R, Jiang Z, Wang H. Clinical Evaluation of an Improved Metagenomic Next-Generation Sequencing Test for the Diagnosis of Bloodstream Infections. Clin Chem. 2021 Aug 5;67(8):1133-1143).
  • To report the potential pathogen, we used a reportable list of clinical pathogens which was set up according to a review of the literature and clinical guidelines and of organisms in the pathogen database referenced in published case reports(Blauwkamp TA, Thair S, Rosen MJ, Blair L, Lindner MS, Vilfan ID, Kawli T, Christians FC, Venkatasubrahmanyam S, Wall GD, Cheung A, Rogers ZN, Meshulam-Simon G, Huijse L, Balakrishnan S, Quinn JV, Hollemon D, Hong DK, Vaughn ML, Kertesz M, Bercovici S, Wilber JC, Yang S. Analytical and clinical validation of a microbial cell-free DNA sequencing test for infectious disease. Nat Microbiol. 2019 Apr;4(4):663-674.)
  • Finally, a composite clinical diagnosis of each patient was adjudicated independently by a committee composed of three independent board-certified infectious disease physicians according to medical history, clinical symptoms, and results of all microbiological tests performed (including staining and microscopy tests, cultures, serology, NTS, and other nucleic acid tests)

Now we have supplemented these reporting rules in the latest version, see line 98-101.

Other questions from the reviewer are responded as follows:

  1. The authors frequently use the term “strain” when they mean “species”. In microbiology (or biology in general), “strain” is applied to genetic variants WITHIN a species. The authors here are intending to refer to different species. And in Supplemental figures 1-4, the X axis label should replace “Number of strains” with “Number of isolates”.)

Response: The axis label "Number of Strains" has been replaced with "Number of isolates" in the supplementary figures 1-4.

  1. How the authors define the “positive rate of HTS” in Table 3 is unclear. For example, on line 110-12 (first paragraph of results), it says that among 67 PDAP samples, “pathogens” were identified in 48 (71.6%) by HTS, and 16 by culture. However, Table 3 indicates a “positive rate of HTS” of 93% for bacterial pathogens, 100% for fungi, and 100% for atypical pathogens. What is the denominator for those calculations? What independent measure of the presence of those pathogens are the authors comparing HTS and culture to? As far as I can tell, there are only two measures of the presence of pathogens being used here - HTS and culture, so they can only be compared to each other, or to the total number of samples.

Response:

Table 3 mainly wants to express the detection efficacy of the two assays for different species such as bacteria, fungi, atypical pathogens, and parasites. The denominator of table 3 refers to the total number of species detected by the two assays, for example, 93% positive rate of HTS in PDAP means that HTS detected 93% of all bacteria detected by the two assays.

  1. Line 246 - The authors note that the commonly detected lung “pathogens” are “not consistent with previous reports” - what isn’t consistent, and why? 

Response:

Our study showed that the pathogens detected by high-throughput sequencing of pulmonary infections were predominantly Gram-positive (54.9%), followed by fungi (21.6%) and Gram-negative (20.7%), and the most common pathogens were Streptococcus, Candida albicans, Stomatococcus mucilaginosus, and Neisseria. However, previous studies reported that common respiratory pathogens were predominantly Gram-negative bacteria, with Pseudomonas aeruginosa, Streptococcus pneumonia and Klebsiella pneumoniae being the most common pathogens. The possible reason maybe the small size of the included patients with lung infection in this study and the fact that the patients had combined kidney disease and were immunocompromised.

  1. Line 257 - The sequencing approach used in this work (nanopore targeted sequencing, which is based on amplification and sequencing of bacterial 16S rRNA sequences, along with fungal internal spacer sequences) cannot identify “all types of pathogens” - it will explicitly NOT be able to detect viral pathogens. Only a metagenomic approach will detect viruses. 

Response:

We have changed this sentence to” In theory, HTS can detect all types of pathogens in clinical samples except viruses”

  1. Supplemental Table 1 - the right side is completely cut off from the PDF so that the lung data isn’t shown. Also in this table, there are numerous typographical errors and misspellings. For example, “Fingoldi magna” should actually be Finegoldia magna. Staphylococcus epidermidis shows up twice, once as “Staphylococcusepidermidis”. There are numerous other errors, and I don’t know what to make of “dystrophic bacteria” and “hemolytic twins” as “identified pathogens”. Please go through this table carefully.

Response:

The supplemental form has been carefully checked and proofread to correct spelling and typographical errors.

Reviewer 2 Report

The authors herein presented a manuscript dealing with the exploitation of the NGS technique for proper bacterial identification in different patients with diverse infections. They also compared the success rate in the identification of pathogens in canonical and standard procedures related to sample colturing. It is with regret that I must unfortunately underline that such as information are already known because several other papers discussed this aspect (eg Mouraviev 2018;Zahng et al, 2022).

Author Response

We appreciate the insightful comments by the reviewer! In the previous version, we left out the NTS reporting rules for potential pathogens, and now we have supplemented these reporting rules in the latest version, see line 98-101. 

Round 2

Reviewer 1 Report

The authors have made welcome improvements to the manuscript after the first round of review. There are just a couple of suggestions that I still think are important. Unfortunately, the line numbers have been deleted in the revised manuscript, so I can't refer those anymore. 

The very first sentence of the results should read that "potential pathogens were identified..." (my emphasis). I appreciate that the authors have included more criteria for assessing the significance of identified microbes from specimens, and added a fourth criteria in the methods explaining that assessment further. But they do refer to this in the methods as "potential pathogen determination", which is appropriate - please use that phrase more often, rather than just referring to whatever microbes were identified in sequencing as pathogens.  

The Table 3 issue I pointed out could be resolved by including footnote(s) with the table explaining how the values were calculated. 

Their response to the line 246 point about "not being consistent with previous reports" should be included somehow in the text, even if just briefly. I accept their explanation, but this won't help the reader if they don't explain in the actual text. 

Thank you to the authors for their responsiveness! 

Author Response

Thank you very much for your patience and suggestive guidance in the whole reviewing process. Below are our responses to the comments raised in round 2 review:

  1. We fully agree with and appreciate the reviewers' requirements for the preciseness of the wording in the manuscript! We have used the phrase "potential" in the text to define the detected microbes in our study.
  2. We have added a footnote to Table3 to explain how the value is calculated. It read as “The positive rate of indicated method for indicated species was calculated by dividing the number of positive samples detected by HTS or culture method by the number of total positive samples detected by the two methods as the denominator.”
  3. The brief explanation has been included in the text according to your advice. It read “which is not consistent with previous reports that common respiratory pathogens were predominantly Gram-negative bacteria, with Pseudomonas aeruginosa, Streptococcus pneumonia and Klebsiella pneumoniae being the most common pathogens [29, 30]. The possible reason maybe the small size of the included patients with lung infection in this study and the fact that the patients had combined kidney disease and were immune compromised.”

Reviewer 2 Report

  Dear authors,

I'm afraid to inform  you that no substantial changes supporting your manuscript have been introduced.

Author Response

Thank you very much for your advice and comments!